# Neurological and Psychiatric Manifestations of Long COVID-19 and Their [^18^F]FDG PET Findings: A Review

**DOI:** 10.3390/diagnostics13142353

**Published:** 2023-07-12

**Authors:** Rizwanullah Hameed, Anuradha Rosario Bahadur, Shashi Bhushan Singh, Juwairah Sher, Maia Todua, Leah (Mahsa) Moradi, Sanjiv Bastakoti, Maeen Arslan, Hanfa Ajmal, Gha Young Lee, Cyrus Ayubcha, Thomas J. Werner, Abass Alavi, Mona-Elisabeth Revheim

**Affiliations:** 1Kingsbrook Jewish Medical Center, 585 Schenectady Avenue, New York, NY 11203, USA; 602drhameed@gmail.com; 2Interfaith Medical Center, 1545 Atlantic Avenue, New York, NY 11213, USA; 3Kingston Public Hospital, North Street, Kingston 4, Jamaica; abrosario0@gmail.com; 4Department of Radiology, Stanford University School of Medicine, Stanford, CA 94304, USA; drmrshashi@gmail.com; 5Medical University of the Americas, 27 Jackson Road, Suite 302, Devens, MA 0134, USA; j.sher@mua.edu; 6Department of Endocrinology, Tbilisi State Medical University, Vazha-Pshavela Ave. 33, 0186 Tbilisi, Georgia; majjka08@gmail.com; 7Touro University, Touro College of Pharmacy, 3 Times Square, New York, NY 10036, USA; mahsa.moradi@touro.edu; 8Russell Hall Hospital, Dudley DY1 2HQ, UK; sanjiv3111@gmail.com; 9Dartmouth College Hanover, Hanover, NH 03755, USA; maeenarslan@gmail.com; 10College of Public Health, University of South Florida, 4202 E Fowler Ave., Tampa, FL 33620, USA; hanfaajmal@yahoo.com; 11Harvard Medical School, 25 Shattuck St., Boston, MA 02115, USA; ghayounglee@hms.harvard.edu (G.Y.L.); cyrusayubcha@hms.harvard.edu (C.A.); 12Department of Radiology, Hospital of the University of Pennsylvania, 3400 Spruce St., Philadelphia, PA 19104, USA; tom.werner@pennmedicine.upenn.edu (T.J.W.); abass.alavi@pennmedicine.upenn.edu (A.A.); 13The Intervention Center, Rikshospitalet, Division for Technology and Innovation, Oslo University Hospital, 0424 Oslo, Norway; 14Institute of Clinical Medicine, Faculty of Medicine, University of Oslo, 0313 Oslo, Norway

**Keywords:** COVID-19, long COVID-19, neurologic, psychiatric, [^18^F]FDG, positron emission tomography, PET

## Abstract

For more than two years, lingering sequalae of COVID-19 have been extensively investigated. Approximately 10% of individuals infected by COVID-19 have been found to experience long-term symptoms termed “long COVID-19”. The neurological and psychiatric manifestations of long COVID-19 are of particular concern. While pathogenesis remains unclear, emerging imaging studies have begun to better elucidate certain pathological manifestation. Of specific interest is imaging with [^18^F]FDG PET which directly reflects cellular glycolysis often linked to metabolic and inflammatory processes. Seeking to understand the molecular basis of neurological features of long COVID-19, this review encompasses the most recent [^18^F]FDG PET literature in this area.

## 1. Introduction

It is now well known that the consequences of COVID-19, as well as its long-term effects, are multisystemic. Typical symptoms of active COVID-19 consist of fever, cough, diminished ability to perceive taste and smell, fatigue, and dyspnea [1,2,3]. Although pulmonary manifestation are of distinct concern, several studies have revealed the impact of COVID-19 on the neurological system [4,5]. Acute symptoms commonly linked to the nervous system include headache and confusion [5,6]. More severe symptoms include delirium, acute behavioral changes, ischemic stroke, and encephalopathy [7,8]. The neuropsychiatric consequences of the COVID-19 pandemic have been manifested possibly both due to direct interaction with the virus as well as socio-economic consequences (e.g., unemployment, financial losses) and behavioral changes (e.g., isolation, stress, and fear) among many other factors [9]. The occurrence and risk factors of long COVID-19 remain unresolved, especially in young people [10].

Neurological symptoms persisting after recovery from COVID-19 due to the long-term systemic ramifications of COVID-19 are termed long COVID-19, post-acute sequelae of COVID-19 (PASC), or post-COVID-19 condition (PCC). Among other systemic symptoms, neurologic and psychiatric manifestations have been noted to persist after acute infection. The most common neurologic symptoms that have been reported in the post-illness stage include post-traumatic stress disorder (PTSD), depression, fatigue, memory and cognitive impairment, “brain fog”, headache, sensorimotor deficits, and sleep/mood/smell/taste disorders [11,12]. The World Health Organization (WHO) has established a definition of long COVID-19 [13] but a recently published study found its prevalence comparable in infected and noninfected subjects raising questions on the usefulness of this definition [10]. While the pathogenesis of these long-term manifestations remains unclear, molecular imaging techniques such as positron emission tomography (PET) scans have explored the underlying molecular basis of long COVID-19. In this review article, we aim to identify the common neurological and psychiatric manifestations of long COVID-19 and the summative conclusions of [^18^F]-fluorodeoxyglucose ([^18^F]FDG) PET/computed tomography (PET/CT) studies.

## 2. Etiopathogenesis of Neuropsychiatric Manifestations of Long COVID-19

COVID-19 is caused by the RNA virus severe acute respiratory syndrome coronavirus 2 (SARS-CoV-2). To date, the virus has undergone many mutations resulting in divergent variants (Table 1), challenging the global collective efforts of preventing the spread of the virus especially through expeditious vaccine development [14].

SARS-CoV-2 enters tissue via ACE2 receptors which have been shown to be found in multiple organ tissues including the brain. Brain regions expressing these ACE2 receptors include the olfactory system, substantia nigra, motor cortex, ventricles, hippocampi, and regions of the brainstem. ACE2 receptors have also been noted to be found in neuronal cells, astrocytes, and oligodendrocytes [15]. The mechanism of entry involves use of the virus’ surface spike protein with subsequent priming of the spike protein by means of serine protease transmembrane protease serine 2 (TMPRSS2). Routes of neuroinvasion include the olfactory epithelium, transsynaptic transfer, vascular endothelium, and leukocyte migration across the blood brain barrier [16]. Given the complexity of the brain structure and functions, possible effects on function are multifaceted. One study demonstrated that post-mortem COVID-19 brains have shown disruption of astrocyte and microglia function [17]. Astrocytes are key regulators of brain metabolism aiding in memory and oxidative protection. During infections, the process of reactive astrogliosis helps to maintain this brain homeostasis; however, the potential failure of glial cell recovery after COVID-19 infection is a likely promoter of the progression of impaired brain homeostasis resulting in the neuropsychiatric effects described in long COVID-19 [18].

Cytokines, such as IL-1β (interleukin-1β), IL-6 (interleukin-6), and TNF-α (tumor necrosis factor alpha), play crucial roles in the propagation of immune response and inflammation [19]. Upon binding to their respective receptors on target cells, these cytokines initiate downstream signaling cascades that mediate various cellular responses. The signaling cascade may continue even after the cessation of antigen exposure [19]. This may explain why there are ongoing neurological changes even after recovery from acute COVID-19 infection.

## 3. Neurological Manifestations of Long COVID-19

The most frequently reported long COVID-19 symptoms of the central nervous system (CNS) are fatigue, headache, memory loss, cognitive impairment, “brain fog”, smell/taste disorders, and sensorimotor deficits [19]. Other neurological findings include status epilepticus, neuropathies such as Guillain-Barré syndrome, and cranial nerve impairment leading to functional deficits [20]. More diffuse symptoms such as difficulty in concentration, forgetfulness, and dizziness have also been reported [21].

Liu et al. reported that an increased number of patients diagnosed with COVID-19 were complaining of new daily persistent headaches weeks after the resolution of viral symptoms [22]. Additionally, many patients who survived COVID-19 have reported some degree of cognitive impairment identified by cognitive screening tests, with poor performance in areas such as memory, performance, and attention. Hoffer found that 21% of seropositive COVID-19 patients reported cognitive impairments after four months of discharge [23]. In addition, del Brutto O. et al. reported that post-pandemic cognitive function using the Montreal Cognitive Assessment (MoCA) was worse in seropositive individuals than those without COVID-19 in the past [24].

Several studies have highlighted that CNS-associated diseases occurring during acute COVID-19 infection may result in long-lasting symptoms. Diseases in the CNS associated with acute COVID-19 infection comprise encephalopathy, ischemic and hemorrhagic stroke, acute disseminated encephalomyelitis, meningitis, and venous sinus thrombosis [24,25,26]. Other damages include hypoxic brain injury and damage to neurons in the regions of the brain vulnerable to hypoxia, which includes the neocortex, cerebellum, and hippocampus [25,26,27]. These diseases and injuries could be linked to long-lasting damage and sustained post-infection symptoms as evidenced by Guan et al. in 2020 where they found that 5% of 221 patients with long COVID-19 had experienced an acute ischemic stroke during acute infection [28]. Other possible mechanisms proposed for the long-term effects and complications of COVID-19 include direct infection of the CNS by the virus, hyper-inflammatory and hypercoagulable states, and post-infectious immune-mediated processes [29]. The persistent neurological manifestations in post-COVID-19 patients have shown to decrease the quality of life, highlighting the importance of rehabilitation in patients displaying prolonged symptoms after acute COVID-19 infection [30].

## 4. Psychiatric Manifestations of Long COVID-19

In the aftermath of COVID-19, patients have reported significant psychiatric symptoms such as depression, post-traumatic stress disorder, anxiety, and sleep disturbances [31]. Moreover, frontline health care professionals have been at an increased risk of suffering from fatigue and psychological symptoms due to increased workload and the trauma from patient deaths [32,33,34]. The factors leading to these neuropsychiatric effects seem to be multifactorial and include physiological response to the infection, pandemic-related stressors such as restrictions on daily activities, severity of disease, and hospitalization [9,35,36,37]. The full effect and duration of these long-term complications of COVID-19 are still undetermined.

Previous studies have shown that natural disasters and health crises result in long-lasting mental health outcomes, for instance, an increase in the incidence of PTSD, substance abuse, depression, maladaptive behavior, emotional distress, panic disorder, somatoform pain disorder, and other neurodegenerative disorders [34,36,37]. Persistent neuroinflammation due to the direct effects of COVID-19 has been proposed as a mechanism of its long-term effects on the nervous system [38], an important theory to highlight as there is an established link between psychiatric conditions such as depression and systemic inflammation. However, more studies are required to demonstrate the direct impact of COVID-19 on the psychological health of individuals [39,40].

In terms of specific psychiatric symptoms, anxiety has been reported as one of the most frequent symptoms in patients during and after the acute phase of the infection, in both symptomatic and asymptomatic individuals. The severity of the anxiety depends on multiple psychosocial and biological factors such as age, gender, comorbidities, lifestyle, and the presence of seemingly unrelated symptoms such as hypogeusia and anosmia [41]. Hence, it is essential to assess and monitor psychiatric symptoms in post-COVID-19 patients. Appropriate psychological interventions, such as awareness programs, should be carried out to address the psychosocial and mental well-being of patients and prevent further psychiatric morbidities.

It is important to note that the neurologic and psychiatric symptoms of long COVID-19 also occur in mild or initially asymptomatic COVID-19 cases. Therefore, it is pertinent that COVID-19 workup involves accurate assessment tools that aid in the diagnosis of long COVID-19 and can identify risks of long-term progression or recovery. Molecular imaging procedures, particularly PET scans, have shown great potential to identify the associated cortical structures involved in persistent neurological and psychiatric symptoms, and therefore, may be a promising tool to predict the risk of long COVID-19 and for the monitoring of the course of the condition.

## 5. PET Findings in the Brain in Patients with Long COVID-19

Various PET studies have investigated COVID-19 patients to assess the effects of the disease on the brain. Early studies have shown hypometabolism of the radiotracer [^18^F]FDG, a fluorine-18 radiolabeled glucose analog, among COVID-19 patients [42]. Specific areas in the brain that showed decreased uptake include the prefrontal and orbitofrontal cortices, the bilateral gyrus rectus, and the cerebellar vermis. Several studies have since been carried out demonstrating similar findings, specifically among long COVID-19 patients. These findings were supported by the 2022 study by Verger et. al. [43], where 143 COVID-19 patients were assessed using [^18^F]FDG PET at 10.9 (±4.8) months from symptom onset. The study found that 21% of scans demonstrated mild to moderately affected brains and 26% of scans showed severe effects from COVID-19 infection [43] based on the visual interpretation analysis of the previously reported long COVID-19 hypometabolic pattern [44]. Brain regions that were affected included the fronto-basal area, limbic and paralimbic areas, and cerebellum. Figure 1 depicts different patients from three different medical centers separated based on the severity of COVID-19 infection and their resulting [^18^F]FDG PET. The regions affected correspond to common complaints from long COVID-19 patients such as taste loss, impaired balance, memory loss, and emotional disturbances [43].

Similarly, one case report by Guedj et al. examined two post-COVID-19 patients and their [^18^F]FDG PET scans [45]. It demonstrated hypometabolism in both patients’ right olfactory/rectal gyrus, but more notably in the patient who experienced prolonged anosmia 4 weeks post-infection and post-COVID-19 cognitive impairment. In addition, the second patient who was experiencing a lower extremity pain syndrome but no anosmia also showed hypometabolism extensively in the bilateral limbic system, right pre- and post-central gyrus, right superior temporal gyrus, cerebellum, and brainstem components (pons and medulla) (Figure 2 and Figure 3) [45].

This was echoed by another case series by the same authors looking at 35 long COVID-19 patients demonstrating reduced [^18^F]FDG metabolism in the olfactory gyrus, associated limbic and paralimbic regions, brainstem, and the cerebellum (Figure 4) [44].

In addition, a case–control study examining long COVID-19 patients showed regions of hypometabolism that correlated with certain long COVID-19 symptoms such as persistent loss of smell/taste (parahippocampal gyri, orbitofrontal cortex) and fatigue (parahippocampal gyrus on right side (Brodmann area 30), substantia nigra in the brainstem, and the thalamus bilaterally [46].

These results were further strengthened by Donegani et al., who confirmed [^18^F]FDG hypometabolism in the left insula as well as bilateral parahippocampus and fusiform gyri among 19 long COVID-19 patients with the sole complaint of hyposmia that was objectively using an olfaction test up to approximately 12 weeks post-COVID-19 [47].

These anatomical brain findings across studies strengthen the theory that the effects of the COVID-19 virus on the nervous system extend from the olfactory bulb and subsequently spread to adjacent and deeper brain structures. These areas are recognized to be implicated in neuropsychiatric illnesses as well as neurodegenerative conditions, which can predispose these patients to said conditions [45]. Complex interconnections from the olfactory areas to other brain regions such as the frontal and limbic areas (hippocampal) play an essential part in exchanging information as part of olfactory memory [47]. Other multiple interconnections aid in multisensory integrations; therefore, these findings indicate valid correlations between long COVID-19 neurologic symptoms and molecular imaging.

Notably, although details on long COVID-19 in the pediatric population are developing, PET hypometabolism has also been demonstrated in pediatric patients after COVID-19 recovery, with similar molecular imaging results seen in adult long COVID-19 patients. Morand et al. published a case series of seven children approximately 5 months post-COVID-19, where PET imaging showed hypometabolism in the olfactory gyrus, medial temporal lobes, pons, and cerebellum (Figure 5) [48].

These findings resemble patterns seen in adult long COVID-19 patients, highlighting the similarity of viral pathology in the brain irrespective of COVID-19 severity or age. Such a pattern was reiterated by Cocciolillo et al. who found orbitofrontal cortex hypometabolism via PET/CT in three pediatric long COVID-19 patients with symptoms of prolonged smell disturbance, issues with short-term memory, concentration difficulties, and headache [49]. In this study, following a visual examination of the patient images, regional [^18^F]FDG PET hypometabolism was identified utilizing a voxel-based comparison with a software package, namely Statistical Parametric Mapping version 8 (SPM8), developed by the Wellcome Department of Cognitive Neurology based in London[49].

One retrospective study examining [^18^F]FDG PET/resting-state functional MRI imaging (rsfMRI) of 13 long COVID-19 patients also demonstrated various regions of hypometabolism, reinforcing the ongoing evidence. This included the frontal, temporal, parietal, occipital lobes, and the thalamus. The parietal and temporal lobes were most frequently affected among the patients. These findings correlated with multiple neurological symptoms including complaints of loss of taste and smell, anxiety, depression, negative mood, fatigue, and myalgia. Mild cognitive impairment was also noted. Additionally, rsfMRI showed atypical connectivity of the brain, which corresponds with the [^18^F]FDG PET findings [50].

Similarly, Hugon et al. reported hypometabolic findings in the brainstem, particularly the pons, in three long COVID-19 cases experiencing a neurologic decline. These patients were experiencing brain fog, memory impairments, and abnormal executive functioning which correlated with the affected brain region. The second case displayed an additional symptom of language deficit correlating with hypometabolism seen in the left parietal lobe, precuneus, and medial temporal lobe. Similarly, the third case displayed an additional symptom of gait instability had hypometabolism in the anterior/posterior cingulate cortex and orbitofrontal cortex (medial part). Notably, two of the three patients were considered mild non-hospitalized COVID-19 cases, and their MRIs were normal in all cases [51].

Most recently, Gorhringer et al. studied 28 non-hospitalized outpatients who were diagnosed with long COVID-19 and considered to have brain involvement. All the long COVID-19 patients in the study displayed neurological symptoms such as cognitive (100%) and language disorders (39%), and headache (46%); with an average MoCA score of 25.9 ± 2.7. Depressive and anxiety symptoms were also noted in 43% of the study group. [^18^F]FDG PET was carried out approximately 16.4 months from symptom onset and revealed diffuse right frontotemporal hypometabolism inclusive of the orbitofrontal cortex and internal temporal areas. These findings also correlated with the patients’ neurological long COVID-19 symptoms. However, in comparison to the studies consisting of previously hospitalized patients, Gorhringer et al. showed no pons/cerebellum hypometabolic involvement [52].

Another case report showed related clinical features and brain [^18^F]FDG PET hypometabolism in a COVID-19 survivor. Areas affected in the COVID-19 survivor 10 months post-hospital discharge were the left temporal lobe, (particularly the superior temporal gyrus) in addition to mild hypometabolism in the surrounding left frontal and parietal regions. There was also mildly asymmetric decreased traces of activity in the left basal ganglia and the thalamus. Correlated neurologic complaints included difficulty in word retrieval, word finding, memory impairments, “brain fog”, poor attention, and impaired executive functioning, which was reported in both cases. The second mild non-hospitalized case was hypometabolic in the parietal and medial temporal lobe, accordant with neurodegenerative dementia. These patients’ MRIs were also unremarkable [53].

Henceforth, these findings suggest that regardless of the initial severity of COVID-19 infection, hypometabolism is similar among long COVID-19 patients, even in those with mild acute infections compared to those with a severe course of the disease. This is reinforced by studies showing the presence of persistent and debilitating long COVID-19 symptoms despite a conventional acute infection [21].

Another study observed seven patients with COVID-19-associated encephalopathy and their [^18^F]FDG PET imaging results at acute and post-COVID-19 phases [54]. It showed consistent hypometabolism of the frontal cortex, insula, anterior cingulate, and caudate nucleus in the acute phase, followed by clinical improvement with the presence of mild residual hypometabolism in the prefrontal, insular, and subcortical areas at six months post-COVID-19 (Figure 6). This coincides with findings of previous studies and demonstrates the lingering effects of post-COVID-19, most notably cognitive, emotional, attention, executive impairments, and anxiety/depression symptoms of varying severity [54].

As evidence on the long-term effects of COVID-19 continues to emerge, it is still being determined whether the imaging findings of cortical hypometabolism are reversible or long-lasting. One prospective study by Blazhenets et al. evaluated eight COVID-19 patients both in the subacute phase and six months post-symptom onset using [^18^F]FDG PET and correlating Montreal Cognitive Assessment (MoCA) scores [55]. They reported that the initial hypometabolism seen in the frontoparietal and temporal regions in the subacute phase decreased at the 6-month follow-up with an improvement in cognition. However, residual neocortical hypometabolism still exists in those recovering patients at follow-up compared to uninfected patients. Additionally, the general performance on the MoCA was below average and fell within the range indicative of mild cognitive impairment. This indicates that full recovery is still uncertain. It is important to note that only half of the patients evaluated at follow-up were still experiencing at least one self-reported cognitive symptom (Figure 7) [55].

In contrast to the above findings, Dressing et al. found no pathologic abnormality in their subset of 14 long COVID-19 patients who underwent [^18^F]FDG PET scans [56]. Moreso, even patients with mildly abnormal MoCA scores showed no [^18^F]FDG differences compared to the controls. Overall, in the entire prospective cohort, cognitive function testing only demonstrated minor impairments with unrevealing PET reports, which implies that neuronal causes may be variable with a possible relation to the predominant symptom of fatigue. However, it must be noted that this study is limited by a small cohort and its lack of findings is not conclusive. Table 2 summarizes major findings from PET studies in long COVID-19 included in this article.

## 6. Conclusions

[^18^F]FDG PET imaging studies of individuals with neurological and psychiatric sequalae of long COVID-19 have shown notable hypometabolism in many regions of the brain. These include the prefrontal and orbitofrontal cortices, the bilateral gyrus rectus, and the cerebellar vermis, among many other structures in the brain. Moreover, studies have demonstrated that hypometabolism in the right olfactory/rectal gyrus is associated with anosmia, parahippocampal gyri, and orbitofrontal cortex with loss of smell and taste, and brainstem with memory impairment and abnormal executive functioning. Similarly, hypometabolism in the frontal cortex, anterior cingulate, insula, caudate nucleus, prefrontal, insular, and subcortical areas have been associated with symptoms related to cognition, attention, emotion, executive impairments, and anxiety/depression. Studies have indicated that regardless of the initial severity of COVID-19 infection, PET hypometabolism findings could be similar among patients with mild, moderate, and severe acute infections. Additionally, pediatric patients have demonstrated PET hypometabolism in patterns similar to adult patients with long COVID-19; however, studies are limited and require further exploration. It is still to be determined whether the imaging findings of PET hypometabolism are reversible or long-lasting past a 6-month period. Thus, [^18^F]FDG PET has shown potential in reflecting sequelae progression, persistence, or recovery among long COVID-19 patients. Nonetheless, future research is necessary to provide more robust support for the presented findings. Studies should in the future include longitudinal and precise documentation of neurological and psychiatric symptomology in larger cohorts. This may better elucidate the association between the long-term neurologic effects of COVID-19 and its structural and biological effects on the brain.

## Figures and Tables

**Figure 1 diagnostics-13-02353-f001:**
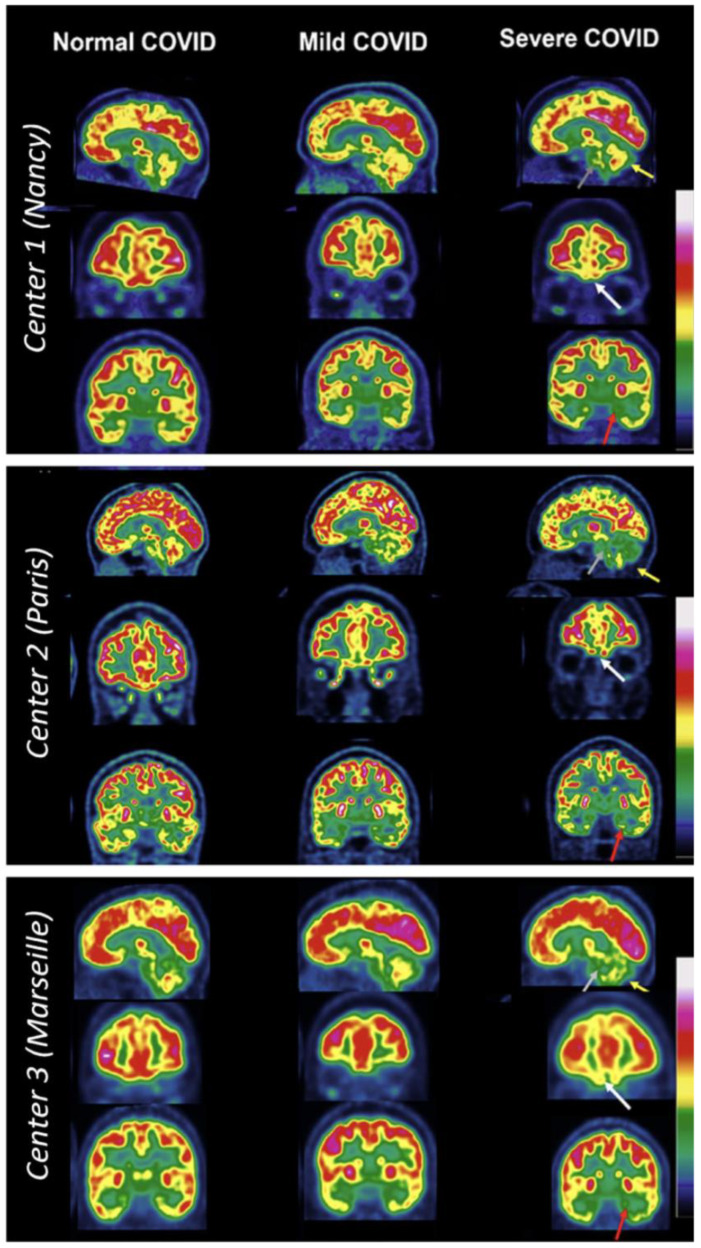
All sets of figures depict [^18^F]FDG PET brain scans of long COVID-19 patients from three different French nuclear medicine centers (depicted in 3 boxes: Nancy, Paris, and Marseilles). These images were captured ranging from 168 days to 545 days after onset of COVID-19. The entire left column depicts normal PET scans, the middle column depicts mild-to-moderate PET hypometabolism, and the right column depicts severe long COVID-19 hypometabolic patterns. The first column with normal PET pattern from top to the bottom includes a 44-year-old woman (with dyspnea and tachycardia), a 35-year-old woman (with persistent asthenia, headache, polyarthralgia, sleep issues, and memory impairment), and a 50-year-old woman (with headache, memory impairment, and anosmia). The middle column consists of images with mild-to-moderate PET hypometabolism, from top to bottom, include a 41-year-old man (with asthenia, dyspnea, and cognitive issues), a 52-year-old woman (with cognitive impairment, asthenia, insomnia, and muscle weakness) and a 55-year-old woman (with memory concerns, headaches, and dyspnea). The right column shows severe PET hypometabolism. From top to bottom, a 37-year-old woman (with dyspnea, orthostatic hypotension, excessive sleep, fever, and cognitive issues), a 38-year-old woman (with memory problems, dizziness, asthenia, polyarthralgia, and myalgia) and a 20-year-old woman (with asthenia, headache, and impairment of memory and concentration). Severe hypometabolic patterns are indicated by arrows: white arrows show the fronto-orbital olfactory regions, red arrows show the other limbic/paralimbic regions, grey for the pons, and yellow for the cerebellum. Reproduced with permission from reference Verger et al. [43].

**Figure 2 diagnostics-13-02353-f002:**
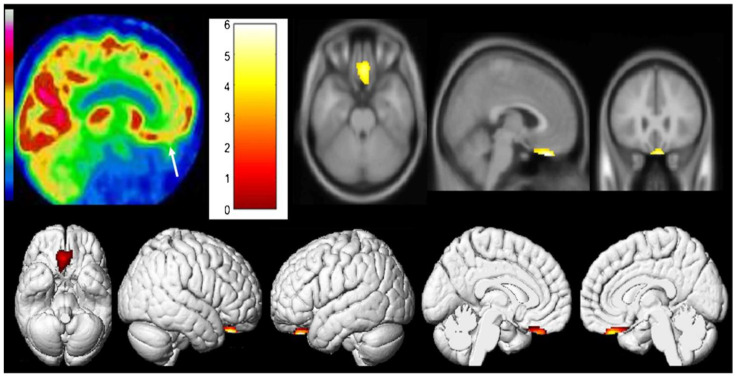
[^18^F]FDG PET hypometabolism in the brain of first patient. Reduced [^18^F]FDG metabolism in the olfactory/rectal gyrus was observed on both sides (indicated by white arrow) and was confirmed by whole-brain voxel-based SPM8 analysis compared to healthy controls (*p* voxel < 0.001; *p*-cluster < 0.05; uncorrected). Reproduced with permission from reference Guedj et al. [45].

**Figure 3 diagnostics-13-02353-f003:**
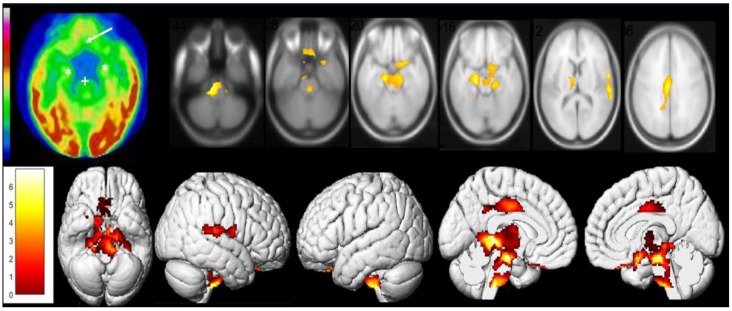
[^18^F]FDG PET hypometabolism in the brain of the second patient. Reduced [^18^F]FDG metabolism in the olfactory and rectal gyrus (indicated by white arrow), medial temporal lobe (indicated by white *), and brainstem (indicated by white +) were observed. The findings were confirmed with whole-brain voxel-based SPM8 analysis compared to healthy controls (*p* voxel < 0.001; *p*-cluster < 0.05; uncorrected). Additionally, other areas of reduced [^18^F]FDG metabolism identified were pre- and post-central gyrus on the right side, superior temporal gyrus on the right side, thalamus on both sides, hypothalamus, and the cerebellum. Reproduced with permission from reference Guedj et al. [45].

**Figure 4 diagnostics-13-02353-f004:**
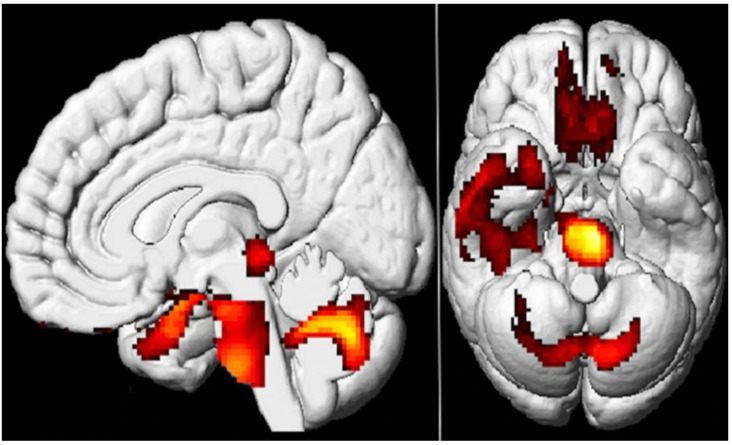
The study examined brain metabolism using [^18^F]FDG PET scans in patients with long COVID-19 and compared them with healthy control. The results showed that the patients with long COVID-19 had lower [^18^F]FDG metabolic activity in specific brain regions such as bilateral rectal and orbital gyrus, including the olfactory gyrus; the right temporal lobe, which includes the amygdala and hippocampus and extends to the right thalamus; the bilateral pons and medulla in the brainstem; and the bilateral cerebellum (colored regions). The study employed SPM8 3D rendering to visualize the findings (*p*-voxel < 0.001 uncorrected and *p*-cluster < 0.05 FWE-corrected). Reproduced with permission from reference [44].

**Figure 5 diagnostics-13-02353-f005:**
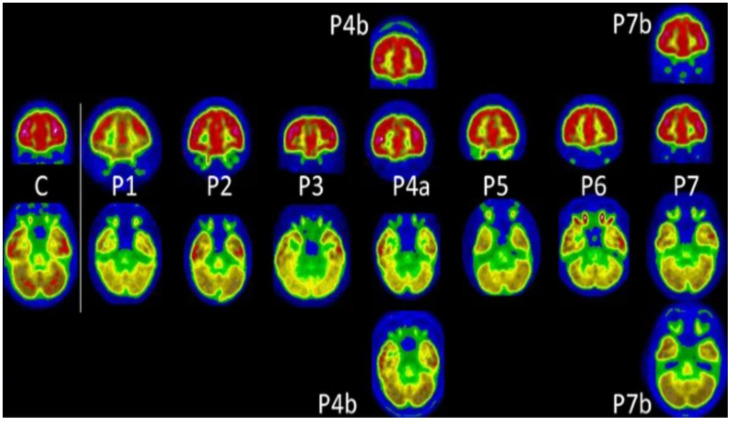
In this study, [^18^F]FDG PET scans were performed on seven pediatric patients (P1 to P7), with follow-up scans conducted for two of them (P4b and P7b). These patients’ main complaints were fatigue, memory issues, and cognitive impairment. A PET scan depicting normal glucose metabolism in a 10-year-old child (C) was also included for reference. The findings revealed reduced metabolic activity in the olfactory regions for children P, 3, 4, 5, and 6; in the temporal regions for children P1, 3, 4, 5, 6, and 7; in the brainstem for children P1, 3, 4, 5, 6, and 7; and in the cerebellum for all the children. During follow-up [^18^F]FDG PET, brain metabolism had improved, especially in the brainstem. Reproduced with permission from reference [48].

**Figure 6 diagnostics-13-02353-f006:**
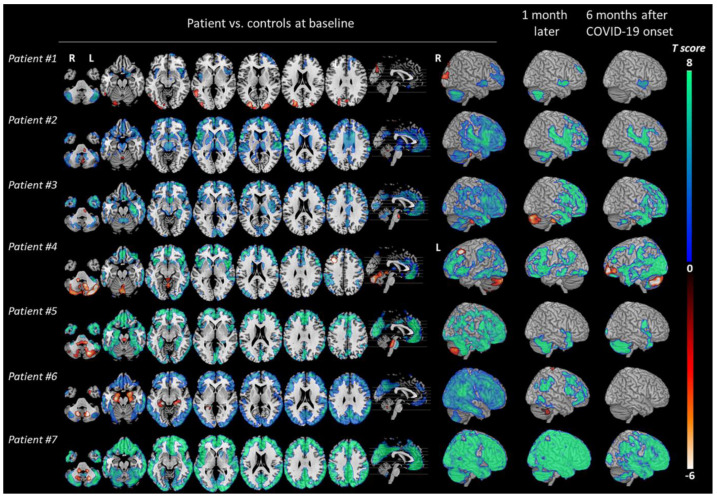
The study examined changes in brain metabolism in COVID-19 patients compared to healthy controls (total of 32 individuals) at different time points: during the acute phase, one month, and six months after the onset of COVID-19. Statistical parametric mapping (SPM) T maps were created for each patient (#1 to #7), displaying hypermetabolism (hot color scale) and hypometabolism (cool color scale) in axial orthogonal views using neurological convention (right side of the image corresponds to the patient’s right). R, right; L, left. The 3D rendering showed the right hemisphere for all patients, except for patient #4, who exhibited hypometabolism in the left frontal cortex due to a focal seizure. Reproduced with permission from reference [54].

**Figure 7 diagnostics-13-02353-f007:**
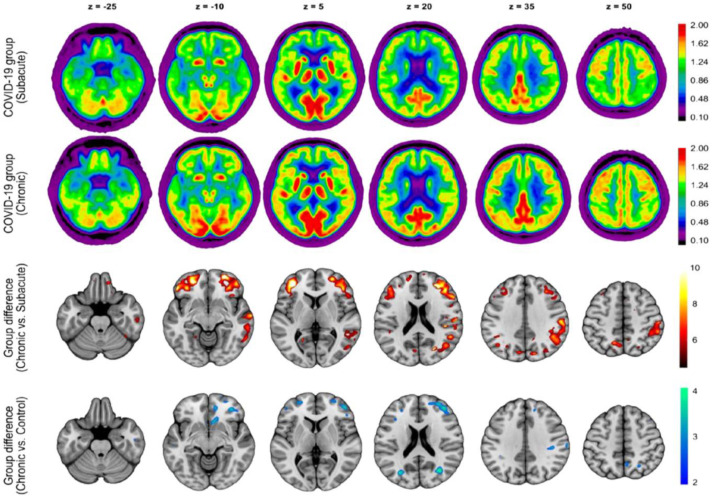
The study conducted a group analysis of [^18^F]FDG PET scans in COVID-19 patients at the subacute and chronic stages. The first and second rows display transaxial sections of the group-averaged, spatially normalized [^18^F]FDG PET scans of eight patients who initially required inpatient treatment for non-neurologic complications. The third row presents regions with significant increases in normalized [^18^F]FDG uptake in COVID-19 patients at the chronic stage compared to the subacute stage, as determined by statistical parametric mapping analysis (paired *t*-test, *p* < 0.01, false-discovery rate corrected). The fourth row depicts regions that still exhibit significant decreases in normalized [^18^F]FDG uptake in COVID-19 patients at the chronic stage compared to an age-matched control group (2-sample *t*-test, *p* < 0.005). The statistical parametric mapping (SPM) 12 t-values are color coded and overlaid onto an MRI template. The images are presented in neurologic orientation, i.e., the left side corresponds to the patient’s left and vice versa. The numbers on the images denote the axial (z) position in millimeters. Reproduced with permission from reference [55].

**Table 1 diagnostics-13-02353-t001:** Variants of SARS-CoV-2 and their origin.

Variant	Year and Country of Origin Reported
Alpha (B.1.1.7)	December 2020 (UK)
Beta (B.1.351)	December 2020 (South Africa)
Gamma (P.1)	January 2021 (Brazil)
Delta (B.1.617.2)	December 2020 (India)
Omicron (B1.1.529)	November 2021 (South Africa)

**Table 2 diagnostics-13-02353-t002:** Summary of [^18^F]FDG PET findings in studies included in this review.

Study	PET Findings ([^18^F]FDG PET Hypometabolism): Main Structures Affected	Correlated Neuropsychiatric Clinical Findings	Days Post-COVID	Study Method/Sample Size
Verger et al. [43]	Fronto-orbital and olfactory regions, limbic and paralimbic regions (amygdalae, hippocampal, and parahippocampal regions), brainstem, and cerebellum	Cognitive impairment, dysexecutive symptoms, memory difficulties, dizziness, limb paraesthesia, and memory/concentration impairment	10.9 months (approximately)	Retrospective, multicenter descriptive study/143 subjects (98 females and 45 males)
Guedj et al. [45]	Olfactory/rectal gyrus (bilateral), medial temporal lobe, brainstem, right pre-/post-central gyrus, the right superior temporal gyrus, bilateral thalamus, hypothalamus, and cerebellum	(Case 1) memory impairment. (Case 2) sensitive lower leg crushing sensation alternating in right/left toes	Not specified	Case reports/2 subjects (both were males and had history of hospitalization)
Sollini et al. [46]	Parahippocampal gyri (Brodmann areas 27 and 36) and orbitofrontal cortex (gyrus rectus, BA 11) on both hemispheres, right parahippocampal gyrus (Brodmann area 30), the brainstem (substantia nigra), and the thalamus of both hemispheres	Anosmia/ageusia and persistent fatigue	132 ± 31 days after diagnosis	Prospective observational case–control/13 subjects (8 males and 5 females)
Blazhenets et al. [55]	Frontoparietal and temporal regions	Mild cognitive impairment (assessed using MoCA)	6 months (approximately)	Prospective cohort/8 subjects (6 males and 2 females)
Kiatkittikul et al. [50]	Parietal lobe, temporal lobe, frontal lobe, occipital lobe, and thalamus	Fatigue/myalgia, negative mood, depression, anxiety, mild cognitive impairment, and loss of taste and smell	>28 days	Retrospective descriptive/13 subjects (6 males and 7 females: 2 had severe COVID-19, presumably hospitalized)
Dressing et al. [56]	None	Impaired attention, memory, and multitasking abilities, word-finding difficulties, and fatigue	197.9 ± 61.1 days	Prospective cohort/31 subjects (11 males and 20 females)
Hugon et al. [51]	Brainstem, particularly the pons (all cases). The left parietal, precuneus and medial temporal lobe (case 2). Anterior and posterior cingulate cortex, and the medial part of the orbitofrontal cortex (case 3)	Brain fog with memory impairments and abnormal executive functioning (all cases), language deficits (case 2), instability with gait (case 3)	Specific not given	Case series/3 subjects (2 males and 1 female; 1 of them had history of hospitalization)
Yu et al. [53]	Left temporal lobe, in particular, superior temporal gyrus, and mild hypometabolism in the adjacent left frontal and parietal regions with mildly asymmetric decreased tracer activity in the left basal ganglia and thalamus (case 1) and parietal and mesial temporal lobe (case 2)	Difficulty in word retrieval, word finding, memory impairments, “brain fog”, poor attention, and impaired executive functioning	5 months and 10 months	Case report/2 subjects (both females; only one of them had history of hospitalization)
Donegani et al. [47]	Relative hypometabolism was demonstrated in bilateral parahippocampal and fusiform gyri and in left insula	Hyposmia	4 and 12 weeks	Prospective cohort/14 subjects (7 males and 7 females)
Morand et al. [48]	Bilateral medial temporal lobes, brainstem and cerebellum, and also the right olfactory gyrus	Fatigue and cognitive impairment such as memory and concentration difficulties	Approximately 5 months	Retrospective/7 pediatric subjects (1 male and 6 females)
Cocciolillo et al. [49]	Left orbitofrontal region	Chronic olfactory dysfunction (1 of 3 cases), short-term memory problems, difficulty in concentrating and headaches (2 of 3 cases)	Approximately 3 months	Observational–case series/3 subjects (1 male and 2 females; all 3 non-hospitalized)
Guedj et al. [44]	Bilateral rectal/orbital gyrus, including the olfactory gyrus; the right temporal lobe, including the amygdala and the hippocampus, extending to the right thalamus; the bilateral pons/medulla brainstem; the bilateral cerebellum	Shortness of breath, chest pain, muscular pain, memory cognitive complaints, insomnia, hyposmia/anosmia, dysgeusia/ageusia	Approximately 97 days	Retrospective/35 subjects (15 males and 20 females; 12 had history of hospitalization in the ICU)
Kas et al. [54]	Prefrontal, insular and subcortical	Cognitive and emotional disorders of varying severity remained with attention/executive disabilities and anxio-depressive symptoms	Approximately 6 months	Observational- prospective/7 subjects (4 males and 3 females; 2 had history of hospitalization)

## Data Availability

Not applicable.

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
