# Peer review of "Neurological and Psychiatric Manifestations of Long COVID-19 and Their [18F]FDG PET Findings: A Review"

_diagnostics, 2023, doi:10.3390/diagnostics13142353_

Round 1

Reviewer 1 Report

the review is valuable and adding more statistical comparison will support the results

Author Response

Thank you for the appreciation and suggestions. We have included statistical comparison from the available literature when possible. However, we are afraid that we found relative scarcity of literature with detailed statistical comparison related to our topic

Reviewer 2 Report

The current article titled “Neurological and psychiatric manifestations of long COVID-19 and their [18F]FDG PET findings: A review” Ref: diagnostics-2438666, discuss an important subject with high understandable langue. It should be recommended for publication after minor changes.

The conclusion section should be modified highlighting the most important observations of the studies mentioned through the article and summarize the future studies needed.

Author Response

The conclusion has been modified to highlight the most important observations of the studies included in the article. We have also summarized the future studies needed

Reviewer 3 Report

The manuscript entitled "Neurological and psychiatric manifestations of long COVID-19 2 and their [18F]FDG PET findings: A review" is well written article in which the authors highlighted the molecular basis of neurological features after long COVID-19. Specifically, the authors synthesize their findings and interpretation highlighting the latest development in the field including use of [18F]FDG PET that could be an index of glycolysis and inflammation.

Line 7-30: Please see the journal guidelines if the email address for each of the authors are needed.

Line 47: Please discuss which virus entity causes Covid-19? I would also introduce/summarize the reader about different type of variants or available molecular structure in the Box form (Box 1). Somewhere in your discussion, it could helpful if you could address the below queries.

a) The route of exposure of Covid-19 virus strain, their incubation period in that tissue, how COVID-19 virus strain enters the BBB to affect the brain functions etc.  To maintain homeostasis, there is protective antioxidant mechanism (See PMID: 35011559) in the body. It would provide more detail understanding how brain homeostasis get perturbed.

b) I understand there could be limitation in PET imaging studies to identify the origin (first appearance), propagation and subsequent development of neurological disorder. It would be helpful to the readers if you could cite few articles that pinpoints which brain compartment are affected first by Covid-19 (what technology they used to reach to this evidence if available) and how it further spread globally into the brain networks to bring the highest level of behavioral level/psychiatric problems.

Line 129: The authors should name few of the cytokines involved in neurological disorders/neuropsychiatric disorders or alternatively they could highlight (at least) the downstream signaling cascade. Making, interpretation of the patient's data coming from long Covid-19 into the general accepted molecular mechanism can strengthened the article prospectus. As the development of the neuropsychiatric disorders would not happen out of blue, it must have followed some signaling cascade. So, the common molecular mechanism follows similar trend whether it toxicant, stress or foreign antigens (See PMID: 36768596). This could explain how the cessation of an antigen exposure could still keep signaling cascade ongoing once it is initiated.

Figure 1-7: The figures look very interesting. Given that number of the subject is very small to quantify the results. However, authors have used the paired t-test. If possible, I suggest showing the quantifiable significant results in bar graph (box plot/pie-diagram or any other presentable form). In that way, it will be clearer to the reader.

Overall, the manuscript is interesting can advance the field forward. I encourage the authors to address the issue I have raised.

Author Response

Point 1: The manuscript entitled "Neurological and psychiatric manifestations of long COVID-19 and their [18F]FDG PET findings: A review" is well written article in which the authors highlighted the molecular basis of neurological features after long COVID-19. Specifically, the authors synthesize their findings and interpretation highlighting the latest development in the field including use of [18F]FDG PET that could be an index of glycolysis and inflammation.

Response 1: Thank you so much for the appreciation.

Point 2: Line 7-30: Please see the journal guidelines if the email address for each of the authors are needed.

Response 2: We have followed the “template” of the journal, and added email addresses for all co-authors.

Point 3: Line 47: Please discuss which virus entity causes Covid-19? I would also introduce/summarize the reader about different type of variants or available molecular structure in the Box form (Box 1).

Response 3: Yes, we have added a table (Table 1) and discussed them in a new section with title, “Etiopathogenesis of Neuropsychiatric Manifestations of Long COVID-19”

Point 4: a) The route of exposure of Covid-19 virus strain, their incubation period in that tissue, how COVID-19 virus strain enters the BBB to affect the brain functions etc.  To maintain homeostasis, there is protective antioxidant mechanism (See PMID: 35011559) in the body. It would provide more detail understanding how brain homeostasis get perturbed.

Response 4: Thank you for pointing this out. We have added a new section with title, “Etiopathogenesis of Neuropsychiatric Manifestations of Long COVID-19”

Point 5: b) I understand there could be limitation in PET imaging studies to identify the origin (first appearance), propagation and subsequent development of neurological disorder. It would be helpful to the readers if you could cite few articles that pinpoints which brain compartment are affected first by Covid-19 (what technology they used to reach to this evidence if available) and how it further spread globally into the brain networks to bring the highest level of behavioral level/psychiatric problems.

Response 5: We have discussed this in the new section with title, “Etiopathogenesis of Neuropsychiatric Manifestations of Long COVID-19”. This has also been explained in more details in the refered papers from Guedj, E. et al. (ref. 44 and 45), and Donegani, M.I. et al. (ref. 47)

Point 6: Line 129: The authors should name few of the cytokines involved in neurological disorders/neuropsychiatric disorders or alternatively they could highlight (at least) the downstream signaling cascade. Making, interpretation of the patient's data coming from long Covid-19 into the general accepted molecular mechanism can strengthened the article prospectus. As the development of the neuropsychiatric disorders would not happen out of blue, it must have followed some signaling cascade. So, the common molecular mechanism follows similar trend whether it toxicant, stress or foreign antigens (See PMID: 36768596). This could explain how the cessation of an antigen exposure could still keep signaling cascade ongoing once it is initiated.

Response 6: Yes, we have included and discussed them in a the new section with title, “Etiopathogenesis of Neuropsychiatric Manifestations of Long COVID-19”

Point 7: Figure 1-7: The figures look very interesting. Given that number of the subject is very small to quantify the results. However, authors have used the paired t-test. If possible, I suggest showing the quantifiable significant results in bar graph (box plot/pie-diagram or any other presentable form). In that way, it will be clearer to the reader.

Response 7: Thank you for your suggestions. We have added the table (Table 2) to the main text which summarizes the major findings in a form presentable to the reader. We are afraid, we could not find the feasibility of a bar graph (box plot/pie-diagram) which could summarize the quantifiable significant results of all the figures comprehensively.

Point 8: Overall, the manuscript is interesting can advance the field forward. I encourage the authors to address the issue I have raised.

Response 8: Thank you so much!

Reviewer 4 Report

This paper aimed to show the neurological and psychiatric manifestations of long-term Covid-19 infection. Also, it should emphasize a special reference to the correlation with PET-MR findings.

However, some changes are needed to improve the manuscript.

1)     The paper has several shortcomings, the main one being that the connection between the clinical picture and PET findings was not highlighted, and regarding the correlation of the neuropsychiatric manifestation of the disease with a specific morphological picture, no significant conclusions were reached.

2)     Figure 1 contains many details related to the images that cannot be followed and that should be simplified.

3)     It is not shown which symptoms the patient from Figure 4 had.

4)     Figure 5. What symptoms did the children have? Hypometabolism was in different regions, and what about the follow-up?

5)     The study by Hugon et al - reference 45 is vaguely and unrelatedly written, so it is difficult to follow the details of the description.

6)     What neurological or psychiatric symptoms were present in Judge Gohringer's patient who describes no involvement of the pons or cerebellum?

7)     The table should be uniform in terms of capital letters, and the thickness of the grid lines inside the table. A column listing the number of subjects in each study should also be put in.

Author Response

Point 1: This paper aimed to show the neurological and psychiatric manifestations of long-term Covid-19 infection. Also, it should emphasize a special reference to the correlation with PET-MR findings.

Response 1: PET/MR is currently available to only limited centers and hence, there has been a relatively scarcity of literature utilizing PET/MR for the evaluation of neurological and psychiatric manifestations of long COVID-19. Hence, our article is lacking PET/MR findings. On the other hand, this also provides uniformity with respect to PET/CT findings in this article. In the absence of PET/MR findings in our article, we have deleted “PET/MRI” from the study objective of our article (last sentence of the introduction) accordingly. Nonetheless, our article includes MR findings at relevant places.

Point 2: The paper has several shortcomings, the main one being that the connection between the clinical picture and PET findings was not highlighted, and regarding the correlation of the neuropsychiatric manifestation of the disease with a specific morphological picture, no significant conclusions were reached.

Response 2: We have updated our conclusion to address this concern.

Point 3: Figure 1 contains many details related to the images that cannot be followed and that should be simplified.

Response 3: The details has been simplified for easier understanding.

Point 4: It is not shown which symptoms the patient from Figure 4 had.

Response 4: Unfortunately, the symptoms were not detailed in the original article.

Point 5: Figure 5. What symptoms did the children have? Hypometabolism was in different regions, and what about the follow-up?

Response 5: The main complaints were fatigue, memory and cognitive impairment (We have added them in the legend). During follow up scans, prior hypometabolism had improved, especially in the brainstem (We have clarified this in the manuscript for easier understanding).

Point 6: The study by Hugon et al - reference 45 is vaguely and unrelatedly written, so it is difficult to follow the details of the description.

Response 6: We have simplified the details for easier understanding.

Point 7: What neurological or psychiatric symptoms were present in Judge Gohringer's patient who describes no involvement of the pons or cerebellum?

Response 7: The details have been added to the main text, as follows: “All the long COVID-19 patients in the study displayed neurological symptoms such as cognitive (100%) and language disorders (39%), and headache (46%); with an average MoCA score of 25.9±2.7. Depressive and anxiety symptoms were also noted in 43% of the study group.”

Point 8: The table should be uniform in terms of capital letters, and the thickness of the grid lines inside the table. A column listing the number of subjects in each study should also be put in.

Response 8: Yes, we have corrected capitalization of letters, thickness of grid lines in the table, and the number of subjects in each study (alongwith gender distribution and history of hospitalization of the patient) has been added in the last column.

Round 2

Reviewer 4 Report

The manuscript was completely revised according to my comments and suggestions.